# Subtypes of Asthma and Cold Weather-Related Respiratory Symptoms

**DOI:** 10.3390/ijerph19148790

**Published:** 2022-07-19

**Authors:** Henna Hyrkäs-Palmu, Maritta S. Jaakkola, Elina M. S. Mäkikyrö, Jouni J. K. Jaakkola

**Affiliations:** Center for Environmental and Respiratory Health Research and Biocenter Oulu, University of Oulu, P.O. Box 5000, FI-90014 Oulu, Finland; henna.hyrkas-palmu@oulu.fi (H.H.-P.); maritta.jaakkola@oulu.fi (M.S.J.); elina.makikyro@oulu.fi (E.M.S.M.)

**Keywords:** asthma control, asthma severity, cold temperature, respiratory symptoms

## Abstract

(1) Poor asthma control increases the occurrence of cold weather-related symptoms among adult asthmatics. We assessed whether the subtype of asthma, taking into account the severity of the asthma, plays a role in these symptoms. (2) We conducted a population-based cross-sectional study of 1995 adult asthmatics (response rate 40.4%) living in northern Finland using a questionnaire that asked about cold weather-related respiratory symptoms including (1) shortness of breath, (2) prolonged cough, (3) wheezing, (4) phlegm production, and (5) chest pain, as well as questions related to the subtype of asthma. For women, the subtypes identified using latent class analysis were: (1) Controlled, mild asthma, (2) Partly controlled, moderate asthma, (3) Uncontrolled, unknown severity, and (4) Uncontrolled, severe asthma, and for men: (1) Controlled, mild asthma, (2) Uncontrolled, unknown severity, and (3) Partly controlled, severe asthma. (3) According to the subtypes of asthma, more severe and more poorly controlled asthma were related to the increased prevalence of cold weather-related respiratory symptoms when compared with those with mild, controlled asthma. This trend was especially clear for wheezing and chest pain. For example, in men, the adjusted prevalence ratio of wheezing was 1.55 (95% CI 1.09–2.19) in uncontrolled asthma with unknown severity and 1.84 (95% CI 1.26–2.71) in partly controlled severe asthma compared with controlled, mild asthma. (4) Our study provides evidence for the influence of subtypes of asthma on experiencing cold weather-related respiratory symptoms. Both women and men reported more cold weather-related symptoms when their asthma was more severe and uncontrolled compared with those who had mild and well-controlled asthma.

## 1. Introduction

The Global Initiative for Asthma (GINA) has defined asthma as a heterogeneous disease characterized by chronic airway inflammation. A typical feature of asthma is the presence of respiratory symptoms in combination with variable airflow obstruction [1]. Rather recent progress in understanding the heterogeneity of asthma has raised the need for more precise knowledge of different asthma phenotypes and for the identification of their typical characteristics.

Recently, several studies have introduced different clustering methods to identify asthma subtypes. These have produced an understanding of the clinical markers that identify the asthma subtypes, or they have combined both clinical markers and questionnaire data into the clustering models to identify relevant combinations [2,3,4,5]. Because in the primary health care setting, such clinical markers or lung function tests may be limited, a different approach utilizing data that are readily available is needed. In addition, in large epidemiological studies, the availability of detailed markers and tests is often limited, so questionnaire-based subtyping of asthma gives a new approach that is likely to be useful for example in primary care and research projects.

Previous studies have reported that cold weather-related symptoms are common among adult populations living in Finland [6,7]. Population-based epidemiological studies have shown that some groups within general populations are more prone to experiencing such cold weather-related symptoms, for example, subjects with chronic lung diseases or cardiovascular diseases [7,8]. Potential relationships between cold weather-related symptoms and asthma severity have not been studied previously, but we have reported that poor asthma control increases the occurrence of cold weather-related symptoms among adult asthmatics [9]. We have also recently presented empirically defined subtypes of asthma [10]. 

The objective of this study was to assess potential relationships between the subtypes of asthma and the occurrence of cold weather-related respiratory symptoms. We hypothesized that subjects with poor asthma control in combination with severe asthma would experience more cold weather-related symptoms than subjects with well-controlled, mild asthma. We tested this hypothesis in the Northern Finnish Asthma Study (NoFAS), in which we first identified these subtypes of asthma.

## 2. Methods

### 2.1. Study Design and Study Population

The NoFAS was initiated in 2012 as a population-based cross-sectional study of 1995 adults with asthma 17–73 years old who were living in northern Finland and were therefor commonly exposed to cold weather. The study was approved by the ethics committee of Oulu University Hospital. The study design and population have been described in more detail in a previous report [9].

### 2.2. Determinant of Interest

The main determinant of interest was asthma subtype based on latent class analysis (LCA). We earlier identified subtypes of asthma in the NoFAS population based on both asthma severity and control. These asthma subtypes were defined utilizing LCA [10]. The variables that were used in the latent class analysis to form the asthma subtypes were asthma control medication use, including bronchodilators, oral corticosteroids, and/or antibiotics during asthma exacerbations and other variables including sick leave days, ER visits, hospital ward stays, acute primary health care visits, and St. George’s Respiratory Questionnaire Total Score, which includes impact score, activity score and symptom score [11]. We found that in the NoFAS study population, the latent classes were more accurate when they were formed for men and women separately [10]. Ours was the first study to examine asthma subtypes among men and women separately in stratified analyses [10]. According to our findings, this is an essential feature in the analyses, especially since the assessment of asthma severity seems to differ substantially between the genders in clinical practice. For women, four subtypes were identified, including (1) Controlled, mild asthma, (2) Partly controlled, moderate asthma, (3) Uncontrolled, unknown severity, and (4) Uncontrolled, severe asthma. For men the subtypes were: (1) Controlled, mild asthma, (2) Uncontrolled, unknown severity, and (3) Partly controlled, severe asthma.

### 2.3. Outcomes

The main outcomes of interest were five cold weather-related respiratory symptoms. These were defined by the following question: “Which asthma symptoms (you can mark more than one symptom) do you experience in cold weather more often than normally/usually: (1) shortness of breath, (2) prolonged cough, (3) wheezing, (4) phlegm production, and (5) chest pain?”

### 2.4. Covariates

The covariates were chosen to represent known or potential determinants of cold weather-related respiratory symptoms, which are likely to be related to the subtypes of asthma. The following covariates were fitted in the multivariate model: age, body mass index (BMI), personal smoking, exposure to secondhand smoke (SHS), cohabitation, education, chronic obstructive pulmonary disease (COPD), allergic rhinitis, and cardiovascular diseases (hypertension, heart failure and/or coronary artery disease). Age was fitted in five categories; BMI in four categories; and smoking, cohabitation, and education in three.

### 2.5. Statistical Methods

We estimated the relationships between the asthma subtype and the occurrence of cold weather-related respiratory symptoms applying prevalence ratios (PR) and their 95% confidence intervals (95% CI). In the crude analysis, we compared the prevalence of the outcomes in four asthma subtypes among women and in three asthma subtypes among men. These were formed based on a previous study that we reported in 2017 [10]. The multivariate analyses were carried out using the GENMOD procedure in the SAS software (SAS 9.4, SAS Institute, Inc., Cary, NC, USA). The analyses applied Poisson regression using the logarithmic link function, and the PRs were adjusted for the covariates described above.

## 3. Results

### 3.1. Characteristics of the Study Population

Table 1 presents the characteristics of the study population, including variables used as covariates in the regression models. Two thirds of the study population were women (65.3%). The BMI distribution varied slightly between men and women, BMI being under 25 more often among women (39.6%) than among men (31.6%). On the other hand, more women (3.5%) than men (1.8%) were extremely obese. The distribution of education level differed between women and men, women having more high-level education (25.1%) than men (19.9%). In addition, the smoking status varied to some extent between women and men, as women were less frequently current smokers (18.0%) and ex-smokers (25.1%) compared with men (19.2% and 39.2%, respectively). 

Subtypes of asthma differed to some extent between women and men in the latent class analysis, as our previous study showed [10]. Table 2 shows that 40% of women in the NoFAS had the mild, controlled asthma subtype, while 28% of men had this mild, controlled subtype. More than half of the men (57%) had the unknown, uncontrolled subtype.

### 3.2. Cold Weather-Related Symptoms in Relation to the Asthma Subtypes

There was an increasing trend of experiencing more cold weather-related respiratory symptoms when asthma was poorly controlled and severe compared with controlled and mild asthma among both women and men. Figure 1 shows that those who were in the uncontrolled, unknown severity subtype reported fewer cold weather-related symptoms than those who were in the severe, uncontrolled (among women) or severe, partly controlled (among men) subgroups. 

Table 3 shows that among women, severe, uncontrolled asthma was the strongest determinant of cold weather-related symptoms. The effect estimates were the highest for wheezing, phlegm production, and chest pain, with adjusted PRs (aPR) of 2.00, 2.36, and 4.48, respectively. Uncontrolled, unknown severity was also a strong determinant of cold weather-related chest pain among women, with aPR = 2.20.

Among men, severe, partly controlled asthma was the strongest determinant of cold weather-related symptoms except for phlegm production, for which the strongest risk was detected among those with unknown, uncontrolled asthma (Table 4). The effect estimates related to unknown, uncontrolled asthma and severe, partly controlled asthma were the highest for chest pain, with aPRs being 2.09 and 3.84, respectively.

## 4. Discussion

This population-based study showed that according to the subtypes of asthma, more severe and more poorly controlled asthma were related to an increased prevalence of cold weather-related respiratory symptoms when compared with with mild, controlled asthma. This trend was clear especially for wheezing and chest pain, symptoms more specific to bronchial asthma. 

### 4.1. Validity of Results 

Our population-based study of adults with asthma included a wide age range of subjects who had special reimbursement rights for asthma medication, which means that their diagnosis was based on standardized criteria including respiratory symptoms and lung function measurements that were the same for the whole study population. A disadvantage of identifying the asthma study population through the national Social Insurance Institution is that because of the law at the time of conducting data collection, researchers were not allowed to contact potential participants directly, so the questionnaires were distributed to them by the Social Insurance Institution. This may have influenced the response rate (40%), which was lower than in our previous epidemiological study [6], as we could not send reminder letters.

The main determinant of interest was the subtype of asthma, which was identified in our previous study among this study population [10]. The subtypes were based on questionnaire data on asthma severity and control, which may have caused some degree of misclassification. If the reporting of cold weather-related respiratory symptoms was related to the subtype of asthma, it was a source of non-differential misclassification leading to the underestimation of the studied effects. The data collected did not include lung function tests to identify the asthma subtypes. We were able to adjust for several determinants of the cold weather-related respiratory symptoms, including age, BMI, personal smoking, exposure to secondhand smoke (SHS), cohabitation, education, and having at the same time chronic obstructive pulmonary disease (COPD), allergic rhinitis, and/or cardiovascular diseases (hypertension, heart failure, or coronary artery disease). Thus, confounding is not a likely explanation for the observed findings.

### 4.2. Synthesis with Previous Knowledge

The potential relationships between asthma subtypes and experiencing cold weather-related symptoms have not been studied previously. Previous studies have indicated that those with respiratory diseases are more prone to report cold weather-related symptoms than those without such diseases [6,7,8]. We previously reported that poor asthma control, based on a widely used Asthma Control Test score, is related to an increased prevalence of cold weather-related symptoms. In addition, there was an increasing trend of such symptoms in relation to the worsening of asthma control [9]. This study assessed of asthma control and severity of asthma at the same time. The questions for asthma control were based on ACT, and the questions for asthma severity have been reported previously. LCA was used to identify the asthma subtypes, and this study showed that these asthma subtypes act as determinants of different cold weather-related symptoms. Among men, the prevalence of cold weather-related symptoms increased with worsening asthma control and/or the increasing severity of asthma, which is consistent with the previous study, although in this study, men were identified as having only three asthma subtypes. Among women, the symptom prevalence differences between the asthma subtypes were found to be larger than those detected between ACT quartiles in the previous study [9]. 

Experimental studies have reported that facial cooling or inhaling cold air can trigger changes in airways, such as increasing inflammation (measured by markers) and obstruction. Cold air is usually dry, which dries the mucosal membrane when inhaling and can result in sensorineural stimulation [12,13,14].

Recently, there has been increasing interest in identifying subtypes for asthma, attempting to identify suitable methods for producing subtypes that would be relevant for clinical practice. Most previous studies have utilized clinical and questionnaire-based data to identify such respiratory disease subtypes, and they have applied different clustering methods [2,3,15,16,17]. One previous study applied LCA to identify childhood asthma and wheezing phenotypes, and they utilized questionnaire-based data to form the phenotypes [3]. In the present study, we identified subtypes of asthma that were formed based on questionnaire data only, as we thought that such application would be useful for example for larger epidemiological studies and in clinical applications where it is not possible to use extensive laboratory-based measurements. In the literature search, we did not find any previous study applying similar methods for a similar study question, so direct comparison with other studies is not possible. 

## 5. Conclusions

Our study provides evidence for the influence of subtypes of asthma based on asthma severity and asthma control on experiencing cold weather-related respiratory symptoms. Both women and men reported more cold weather-related symptoms when their asthma was more severe and uncontrolled compared with those who had mild and well-controlled asthma. We identified among women four subtypes of asthma and among men three subtypes. Among women, those with moderate, partly controlled asthma experienced more cold weather-related symptoms than those with mild, controlled asthma. The unknown, uncontrolled and severe, uncontrolled asthma patients experienced significantly more of the studied cold weather-related symptoms. Among men, severe, partly controlled asthma had the strongest effect on experiencing cold weather-related symptoms, except for phlegm production. The results of this study are of importance for asthma patients as well as for the health care personnel treating and managing their asthma. These results help in identifying which adult asthma subtypes are most vulnerable to experiencing cold weather-related respiratory symptoms, so that health care personnel can inform their asthma patients about their vulnerability to cold weather-related effects and advise them on how to protect themselves from cold weather-related symptoms. These results are particularly relevant to populations living in the northern hemisphere, where populations are exposed to long, cold, dry winter periods.

## Figures and Tables

**Figure 1 ijerph-19-08790-f001:**
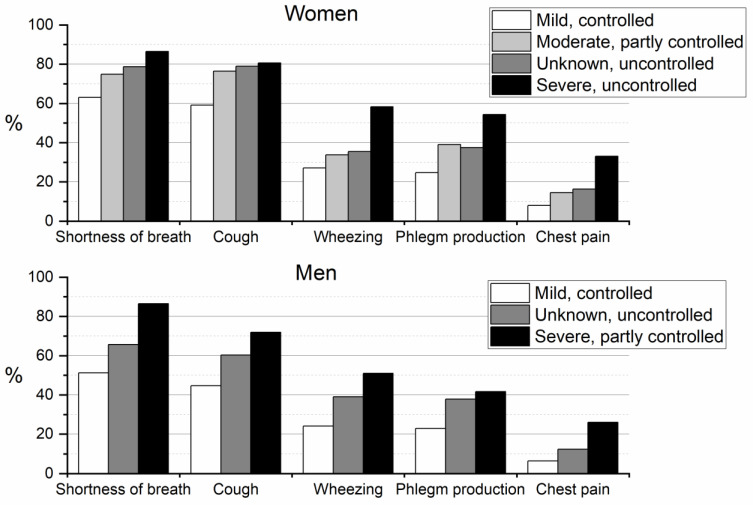
Bar graph of the cold weather-related symptoms according to the asthma subtype.

**Table 1 ijerph-19-08790-t001:** Characteristics of the study population.

	Women N (%)	Men N (%)	Total N (%)
**Total**	1303 (65.31)	692 (34.69)	1995
**Age**			
<30	141 (10.82)	71 (10.26)	212 (10.63)
30–40	177 (13.58)	91 (13.15)	268 (13.43)
40–50	243 (18.65)	111 (16.04)	354 (17.74)
50–60	428 (32.85)	218 (31.50)	646 (32.38)
>60	314 (24.10)	201 (29.05)	515 (25.81)
**Body mass index**			
<25	504 (39.56)	216 (31.58)	720 (36.77)
25–30	420 (32.97)	286 (41.81)	706 (36.06)
30–35	305 (23.94)	170 (24.85)	475 (24.26)
>35	45 (3.53)	12 (1.75)	57 (2.91)
Missing	29	8	37
**Marital status**			
Single	155 (11.91)	121 (17.51)	276 (13.86)
Marriage/Cohabitation	963 (74.02)	498 (72.07)	1461 (73.34)
Divorced, separated or widow	183 (14.07)	72 (10.42)	255 (12.80)
Missing	2	1	3
**Education**			
Low	298 (22.99)	205 (29.71)	503 (25.33)
Medium	673 (51.93)	348 (50.43)	1021 (51.41)
High	325 (25.08)	137 (19.86)	462 (23.33)
Missing	7	2	9
**Smoking**			
Current smoker	231 (17.98)	134 (19.62)	365 (18.55)
Ex-smoker	323 (25.14)	268 (39.24)	591 (30.03)
Never smoked	731 (56.89)	281 (41.14)	1012 (51.42)
Missing	18	9	27

**Table 2 ijerph-19-08790-t002:** The subtypes of asthma among women and men in NoFAS.

	N (%)
**Women**	
Mild, controlled	525 (40.29)
Moderate, partly controlled	287 (22.03)
Unknown, uncontrolled	376 (28.86)
Severe, uncontrolled	115 (8.83)
**Men**	
Mild, controlled	193 (27.89)
Unknown, uncontrolled	395 (57.08)
Severe, partly controlled	104 (15.03)

**Table 3 ijerph-19-08790-t003:** The prevalence and prevalence ratios of cold weather-related respiratory symptoms according to the subtypes of asthma among women, Northern Finnish Asthma Study (2012).

	No. of Subjects (%)	Crude PR (95% CI)	Adjusted PR (95% CI)
**Shortness of breath**			
Mild, controlled	301 (63.10)	1.00	1.00
Moderate, partly controlled	191 (74.90)	1.19 (1.08, 1.31)	1.17 (1.05, 1.29)
Unknown, uncontrolled	269 (78.65)	1.25 (1.14, 1.36)	1.26 (1.14, 1.40)
Severe, uncontrolled	89 (86.41)	1.37 (1.24, 1.52)	1.37 (1.23, 1.53)
**Cough**			
Mild, controlled	282 (59.12)	1.00	1.00
Moderate, partly controlled	195 (76.47)	1.29 (1.17, 1.43)	1.24 (1.12, 1.38)
Unknown, uncontrolled	270 (78.95)	1.34 (1.22, 1.46)	1.34 (1.21, 1.49)
Severe, uncontrolled	83 (80.58)	1.36 (1.21, 1.54)	1.35 (1.19, 1.53)
**Wheezing**			
Mild, controlled	129 (27.04)	1.00	1.00
Moderate, partly controlled	86 (33.73)	1.25 (0.99, 1.56)	1.23 (0.97, 1.55)
Unknown, uncontrolled	121 (35.48)	1.31 (1.07, 1.61)	1.30 (1.02, 1.64)
Severe, uncontrolled	60 (58.25)	2.15 (1.73, 2.68)	2.00 (1.56, 2.57)
**Phlegm production**			
Mild, controlled	118 (24.74)	1.00	1.00
Moderate, partly controlled	99 (38.98)	1.58 (1.27, 1.96)	1.54 (1.22, 1.94)
Unknown, uncontrolled	128 (37.43)	1.51 (1.23, 1.86)	1.60 (1.27, 2.01)
Severe, uncontrolled	56 (54.37)	2.20 (1.74, 2.78)	2.36 (1.83, 3.03)
**Chest pain**			
Mild, controlled	38 (7.98)	1.00	1.00
Moderate, partly controlled	37 (14.57)	1.82 (1.19, 2.79)	1.61 (1.03, 2.53)
Unknown, uncontrolled	56 (16.37)	2.05 (1.39, 3.02)	2.20 (1.40, 3.48)
Severe, uncontrolled	34 (33.01)	4.13 (2.74, 6.24)	4.48 (2.83, 7.09)

**Table 4 ijerph-19-08790-t004:** The prevalence and prevalence ratios of cold weather-related respiratory symptoms according to the subtypes of asthma among men, Northern Finnish Asthma Study (2012).

	No of Subjects and Symptom Prevalence	Crude PR (95% CI)	Adjusted PR (95% CI)
**Shortness of breath**			
Mild, controlled	87 (51.18)	1.00	1.00
Unknown, uncontrolled	222 (65.68)	1.28 (1.09, 1.51)	1.44 (1.18, 1.75)
Severe, partly controlled	83 (86.46)	1.69 (1.43, 2.00)	1.79 (1.46, 2.21)
**Cough**			
Mild, controlled	76 (44.71)	1.00	1.00
Unknown, uncontrolled	204 (60.36)	1.35 (1.12, 1.63)	1.38 (1.10, 1.73)
Severe, partly controlled	69 (71.88)	1.61 (1.30, 1.98)	1.55 (1.20, 2.00)
**Wheezing**			
Mild, controlled	41 (24.12)	1.00	1.00
Unknown, uncontrolled	132 (39.05)	1.62 (1.20, 2.18)	1.55 (1.09, 2.19)
Severe, partly controlled	49 (51.04)	2.12 (1.52, 2.95)	1.84 (1.26, 2.71)
**Phlegm production**			
Mild, controlled	39 (22.94)	1.00	1.00
Unknown, uncontrolled	128 (37.87)	1.65 (1.21, 2.25)	1.82 (1.27, 2.59)
Severe, partly controlled	40 (41.67)	1.82 (1.26, 2.61)	1.75 (1.14, 2.68)
**Chest pain**			
Mild, controlled	11 (6.47)	1.00	1.00
Unknown, uncontrolled	42 (12.43)	1.92 (1.01, 3.63)	2.09 (0.97, 4.54)
Severe, partly controlled	25 (26.04)	4.02 (2.07, 7.81)	3.84 (1.72, 8.57)

## Data Availability

Data will not be shared for reasons of confidentiality.

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
