# Peer review of "Subtypes of Asthma and Cold Weather-Related Respiratory Symptoms"

_ijerph, 2022, doi:10.3390/ijerph19148790_

Round 1

Reviewer 1 Report

The authors have presented a study on the prevalence of asthmatic symptoms in winter on asthmatic male/female people classified into different subtypes of asthma based on questionnaire data in an epidemiological study conducted in Finland. The results show that more severe and more poorly controlled asthma were related to increased prevalence of cold weather-related respiratory symptoms compared to those with mild, controlled asthma. The response from male (3 subtypes) and female (4 subtypes) cohorts are slightly different but the  more severe and uncontrolled asthma have increasing symptoms compared to controlled or mild controlled asthma.

I find the methods used are strong and the analysis was conducted in details. Asthmatic symptoms are usually triggered by either pollens, house mites,... or by weather-related condition (humidity, dampness,..). In the case of winter period, humidity should be a factor to be considered.

Overall, I recommend the manuscript to be accepted for publication with some minor comment or consideration on the humidity issue. Following is also some specific minor comments

(1) Line 22: "unknow" should be "unknown"

(2) Line 58: NoFAS should have full name here rather a few lines below: Northern Finnish Asthma Study (NoFAS)

Author Response

REVIEWER 1

Comment 1

I find the methods used are strong and the analysis was conducted in details. Asthmatic symptoms are usually triggered by either pollens, house mites,... or by weather-related condition (humidity, dampness,..). In the case of winter period, humidity should be a factor to be considered.

Response 1

Thank you. We have added some discussion of the humidity-related effects, i.e. that the winters in this area are cold with dry air?

(lines 207-208). “Cold air is usually dry which dries the mucosal membrane when inhaling and can result in sensorineural stimulation [12-14].”

Comment 2

Line 22: "unknow" should be "unknown"

Response 2

We have corrected this misspelling.

Comment 3

Line 58: NoFAS should have full name here rather a few lines below: Northern Finnish Asthma Study (NoFAS)

Response 3

We have added the full name of the study (on line 58).

Reviewer 2 Report

This paper mainly investigates the relationship between the subtype of asthma and cold-weather related symptoms in adult patients with asthma. The population-based cross-sectional study with gender stratification was conducted. It has certain research significance. However, there seems to be some discrepancy between the title and content, such as the symptoms related to cold weather (chronic lung disease or cardiovascular disease with documentary evidence). The article describes these five categories: shortness of breath, cough, asthma, sputum and chest pain. These are indeed related, but are they caused by cold weather? Moreover, there are some problems in the text:

 1. Introduction: line 52, for asthma subtype, it is based on asthma control and asthma severity. However, the literature cited above that only considers the control of asthma, and does not discover the severity of asthma. Can relevant literature be introduced to explain, so as to put forward the current research direction.

 2. Methods: line 79, why are there different asthma subtypes based on gender analysis? What is the basis? Line 90, what are the main factors considered by the covariate? The authors should provide specified description and explanation to support these analyses.

 3. Results: line 108, there are few descriptions of the results. For example, there are great differences in gender analysis at the smoking stage, but they are not explained. In line 122, the presentation of the result is also missing.

 4. Discussion: I suggest the authors to improve the Discussion. The discussion needs more detailed information. There will be valuable and interesting findings based the results. Line 174, the authors could add more discussion on the results from literature with different findings?

 5. Conclusions: it is suggested to add a paragraph of its own conclusion on this phenomenon to the summary of each result.

Author Response

REVIEWER 2

Comment 1

However, there seems to be some discrepancy between the title and content, such as the symptoms related to cold weather (chronic lung disease or cardiovascular disease with documentary evidence). The article describes these five categories: shortness of breath, cough, asthma, sputum and chest pain. These are indeed related, but are they caused by cold weather?

Response 1

Thank you for your comments. The question of symptoms has been introduced in Methods and it points out that the question asked particularly about experiencing cold weather -related worsening of respiratory symptoms when the study subjects were in cold air (lines 84-88).

Comment 2

Introduction: line 52, for asthma subtype, it is based on asthma control and asthma severity. However, the literature cited above that only considers the control of asthma and does not discover the severity of asthma. Can relevant literature be introduced to explain, so as to put forward the current research direction.

Response 2

We have included this information onto lines 53-54 by adding a sentence and citing ref. 10, where the subtypes of asthma were introduced. “We have recently presented empirically defined subtypes of asthma [10].”

Comment 3

Methods: line 79, why are there different asthma subtypes based on gender analysis? What is the basis?

Response 3

The basis of having three subtypes for men and four subtypes for women was discussed in our previous study, reference [10]. We have added citation to this reference on line 84 and two new sentences on lines 84-87:

“Ours was the first study to examine asthma subtypes among men and women sepa-rately in stratified analyses [10]. According to our findings this is an essential feature in the analyses, since especially the assessment of asthma severity seems to differ sub-stantially between the genders in clinical practice, so stratified analyses were conducted.”

Comment 4

Line 90, what are the main factors considered by the covariate? The authors should provide specified description and explanation to support these analyses.

Response 4

Covariates were chosen to represent known or potential determinants of cold weather-related respiratory symptoms, which are likely to be related at the same time to the subtypes of asthma.

Comment 5

Results: line 108, there are few descriptions of the results. For example, there are great differences in gender analysis at the smoking stage, but they are not explained.

Response 5

We have added more results of the characteristics of the study population (lines 124-128):

“The distribution of education level differed between women and men, women having more high-level education (25.1%) than men (19.9%). In addition, the smoking status varied to some extent between women and men, as women were less frequently cur-rent smokers (18.0%) and ex-smokers (25.1%) compared to men (19.2% and 39.2%, respectively).”

In this descriptive analysis, our main interest was to see what kind of differences there are according to the subtypes of asthma.

Comment 6

In line 122, the presentation of the result is also missing.

Response 6

We have added some text on results here.

Comment 7

Discussion: I suggest the authors to improve the Discussion. The discussion needs more detailed information. There will be valuable and interesting findings based the results. Line 174, the authors could add more discussion on the results from literature with different findings?

Response 7

On the basis of a systematic search, we have identified all the relevant publications on this topic. This is a limited body of evidence and therefor the discussion is also rather short.

Comment 8

Conclusions: it is suggested to add a paragraph of its own conclusion on this phenomenon to the summary of each result.

Response 8

Following Reviewer 2’s suggestion, we have slightly expanded the conclusion section.

Reviewer 3 Report

The manuscript is well written using data that was adequately collected.

However, the authors need to provide the pathophysiological mechanism of the relationship between cold weather and the asthma subtypes.  

Using the literature, the authors will can help the reader to understand if this study is only relevant to Northern Finland. Or can it be generalized to other countries with cold weather. 

What are the public health implications of their findings? Can this study help save lives among asthmatics? 

Author Response

REVIEWER 3

Comment 1

The manuscript is well written using data that was adequately collected. However, the authors need to provide the pathophysiological mechanism of the relationship between cold weather and the asthma subtypes.

Response 1

We have added some discussion of the possible pathophysiological mechanisms on lines 208-211.

Comment 2

Using the literature, the authors will can help the reader to understand if this study is only relevant to Northern Finland. Or can it be generalized to other countries with cold weather.

Response 2

We have added a sentence concerning the relevance of these results (lines 242-243).

“These results are relevant to populations living in the northern hemisphere where populations are exposed to long cold and dry winter periods.”

Comment 3

What are the public health implications of their findings? Can this study help save lives among asthmatics?

Response 3

As a response to Reviewer 3’s comment, we have modified and slightly expanded Conclusion (lines 237-239).

“These results help in identifying which adult asthma subtypes are most vulnerable to experiencing cold weather-related respiratory symptoms, so that health care personnel can inform their asthma patients about their vulnerability to cold weather-related effects and advise on how to protect themselves from cold weather-related symptoms.”

Round 2

Reviewer 2 Report

The authors have revised the manucsript according to my suggestions, and responded all my comments in details. I have no more comments.

Author Response

Reviewer has approved the first revised version as stated below.

We have checked the final version and uploaded today a clean version of the manuscript.